# PAI Materials Synthesized by 4,4′-Diaminodiphenyl ether/2,2′-Bis (trifluoromethyl)-4,4′-diaminophenyl ether and Their Properties

**DOI:** 10.3390/ma14216376

**Published:** 2021-10-25

**Authors:** Haiyang Yang, Duxin Li, Jun Yang, Jin Wang, Shunchang Gan

**Affiliations:** 1State Key Laboratory of Powder Metallurgy, Central South University, Changsha 410083, China; haiyang1983425@sina.com; 2Zhuzhou Times New Material Technology Co., Ltd., Zhuzhou 412007, China; Wangjin@csrzic.com (J.W.); gansc@csrzic.com (S.G.)

**Keywords:** polyamide-imide (PAI), 4,4′-diaminodiphenyl ether, 2,2′-bis (trifluoromethyl)-4,4′-diaminophenylether

## Abstract

In this paper, 4,4′-diaminodiphenyl ether and 2,2′-bis (trifluoromethyl)-4,4′-diaminophenyl ether are selected for molecular structure design, and PAI materials are synthesized by acyl chloride method. 2,2′-bis (trifluoromethyl)-4,4′-diaminophenyl ether has the same main chain structure as 4,4′-diaminodiphenyl ether, but the side chain contains two trifluoromethyl groups, which has high fluorine content. PAI terpolymers were prepared by compounding two diamine monomers, and the effects of trifluoromethyl on heat resistance, friction and wear properties, hydrophobic properties and mechanical properties of PAI materials were studied. The results showed that with the increase of trifluoromethyl content, the Tg of PAI material first increased and then changed little, and the Td_5%_ would decrease and the tensile properties would also decrease. The wear mechanism of PAI varied with the content of trifluoromethyl. With the increase of the amount of fluorinated diamine monomer, the adhesive wear degree of PAI materials gradually increased, and reached the maximum when the molar ratio of the two monomers was 5:5, and then decreased gradually. Different trifluoromethyl content had little effect on friction coefficient, and the friction coefficient increased slightly when the molar ratio of 4,4′-diaminodiphenyl ether to 2,2′-bis (trifluoromethyl)-4,4′-diaminophenyl ether is 1:9. With the increase of trifluoromethyl content, the wear of PAI material would increase. With the increase of the amount of trifluoromethyl, the water absorption of PAI material decreased and the water contact angle increased, which indicated that the hydrophobic property of PAI material was improved. To sum up, the results of this study showed that the introduction of trifluoromethyl into the side chain provided an effective way to prepare PAI materials with low water absorption. Considering the comprehensive properties such as heat resistance, friction and wear, tensile properties, etc., the appropriate addition amount is 10–30%.

## 1. Introduction

Polyimide (PI) is a kind of special engineering material with excellent performance [1,2,3,4]. Polyamide-imide (PAI) is a modified polyimide, which usually has repeated structural units (Figure 1). It can be seen from the molecular structure that PAI contains two structural units, amide bond and imide ring, so PAI has some performance advantages over PA and PI, such as high-temperature resistance, corrosion resistance, friction and wear resistance, excellent mechanical properties, and can be compounded with various substances, etc. PAI is a special engineering plastic with excellent performance, which can be applied in extremely severe working conditions [5,6].

PAI polymer synthesized by 4,4’-diaminodiphenyl ether has better comprehensive properties, and the raw material is cheap, which has better engineering application prospects. 2,2’-bis (trifluoromethyl)-4,4’-diaminophenyl ether has the same main chain structure as 4,4’-diaminodiphenyl ether, and the side chain contains two trifluoromethyl groups, which has high fluorine content. Studies have shown that introducing fluorine or fluorine-containing groups into the molecular structure can reduce the moisture absorption rate of materials [7,8,9], improve the heat resistance [8,10], improve the wear resistance [11,12,13], improve the solubility [14], and so on.

There were few reports onthe effects of fluorine atoms and fluorine-containing groups on PAI. In this paper, 2,2’-bis (trifluoromethyl)-4,4’-diaminophenyl ether and 4,4’-diaminodiphenyl ether were compounded to synthesize PAI by the acyl chloride process [5,15]. The effects of trifluoromethyl on heat resistance, friction and wear resistance, hydrophobicity, and mechanical properties of PAI materials were studied to prepare PAI materials with excellent properties.

## 2. Experimental

### 2.1. Materials

4,4’-diaminodiphenyl ether was analytically pure and purchased from Aladdin (Shanghai, China). 2,2’-bis (trifluoromethyl)-4,4’-diaminophenyl ether was analytically pure and purchased from Changzhou Sunshine Pharmaceutical Technology Co., Ltd. (Changzhou, China). 1,2,4-trimellitic anhydride acyl chloride(TMAc) was purchased from Tokyo Chemical Industry (Tokyo, Japan). Triethylamine as an acid-binding agent and catalyst was purchased from Aladdin (Shanghai, China). Acetic anhydride was analytically pure, used as dehydrating agent, purchased from Aladdin (Shanghai, China). N-methylpyrrolidone(NMP) as aprotic polar solvent was purchased from Aladdin (Shanghai, China).

### 2.2. Preparation of PAI

PAI synthesized by 4,4’-diaminodiphenyl ether and TMAC was named PAI(O). When the molar ratio of 4,4’-diaminodiphenyl ether to 2,2’-bis (trifluoromethyl)-4,4’-diaminophenyl ether is (9:1), (7:3), (5:5), (3:7) and (1:9), the synthesized PAI terpolymers were named PAI-1,PAI-2,PAI-3,PAI-4,PAI-5 in turn.PAI synthesized by 2,2’-bis (trifluoromethyl)-4,4’-diaminophenyl ether and TMAC was named PAI(F).The preparation process was as follows:(1)Polymerization process

Under the protection of nitrogen, NMP was added into a 500 mL four-necked flask, and the amount of NMP was 8 times the total mass of trimellitic anhydride acyl chloride and diamine monomer. Then, 0.05 mol of fluorine-containing diamine monomer was added and stirred at high speed until it was completely dissolved. 0.052 mol (10.9497 g) of trimellitic anhydride acyl chloride was added in batches, the material temperature was controlled at 0–5 °C, and then 0.052 mol (5.2619 g) of acid-binding agent triethylamine was added dropwise. After adding materials, continued to react for 12 h, then raised the temperature, controlled the temperature at 60–65 °C, and continued to react for 4 h. After the above steps were completed, the chemical imidization reagent (mixed solution of 0.0625 mol acetic anhydride and 0.0625 mol triethylamine) was added dropwise, the temperature of imidization process was controlled at 60–65 °C, and the reaction was timed for 12h to ensure the full progress of imidization.

The process flow for synthesizing PAI by reacting different fluorinated diamine monomers with acyl chloride is shown in Figure 1.
(2)Post-processing process

The material was poured into 5 L deionized water, precipitated, mashed, washed with water, and filtered by suction to obtain PAI resin powder. Set the temperature of the blast oven at 100 °C and dry for 12 h to remove the moisture in the materials. The temperature of the vacuum oven was set at 200 °C, and heat treatment was carried out for 1 h to remove the residual solvent in the material.
(3)Sample preparation method

PAI resin was dissolved in NMP solvent to prepare a 20 wt.% PAI solution. PAI solution was cast on the surface of 45# steel sample after sandblasting, and then the friction and wear samples were prepared by temperature-programmed baking. PAI film was obtained by scraping the PAI solution onto a clean glass plate and then baking with programmed temperature, which was used to test its tensile properties, water absorption, and water contact angle. The temperature-programmed process was as follows: it took 10 min for room temperature to rise to 50 °C, and then kept it for 4 h. It took 10 min to raise the temperature from 50 °C to 80 °C and then kept it for 4 h. It took 10 min to raise the temperature from 80 °C to 150 °C and then kept it for 1 h. It took 10 min to raise the temperature from 150 °C to 200 °C and then kept it for 1 h. Finally, naturally cooled them to room temperature.

### 2.3. Characterization

FT-IR analysis. PAI material was measured by Nicolet IS/10 Fourier infrared spectrometer produced by Thermo Nicolet company. The test sample was dried and then mixed with KBr powder for tableting. The samples were tested by the attenuated total reflectance method with a resolution of 2 cm^−1^. The scanning range was 4000–450 cm^−1^, and the number of scans was 32 times.

Molecular weight size and polydispersity index. ISO16014-1:2012 was selected as the detection standard, and the test was carried out with WATERS 1515 gel permeation chromatography. The ambient temperature of the test is 23 ± 1 °C. DMF was selected as the solvent, and 0.1 mol/L LiCl was added to the samples with poor solubility to promote the dissolution and dispersion of the samples, which were filtered with 0.45 μm syringe organic phase filter head. The molecular weight of the sample was tested with polystyrene as the calibration standard and the flow rate was set to 1 mL/min.

Mechanical properties test. Tensile strength and elongation at break of materials were tested on JYW-93 Z100 electronic universal testing machine, the test standard was GB/T1040-2006, the tensile rate was 50 mm/min, the number of test splines was 5, and the arithmetic average was taken.

DSC analysis. Chose DSC1/JYH-86 thermal analyzer of METTLER TOLEDO company in Switzerland to test the glass transition temperature (Tg) of synthetic polymer. During the test, nitrogen was protected, and the heating rate was 10 °C/min.

TG analysis. The thermogravimetric test of materials was carried out on TG209 equipment produced by NETZSCH Company in Germany, and the temperature rise rate was 10 °C/min under the protection of nitrogen.

Water absorption rate test. The water absorption rate test was conducted according to plastics-determination of water absorption, ISO 62:2008. The sample size was (61 ± 1) × (61 ± 1) × (0.1 ± 0.01) mm.

Put the sample in an oven at 50.0 ± 2 °C for drying 24 ± 1 h, then cooled it to room temperature in the dryer, and weighed each sample to the accuracy of 0.1 mg. Put the sample in a container filled with distilled water, controlled the water temperature at 23.0 ± 1.0 °C, soaked it for 24 ± 1 h, took out the sample, wiped off all the water on the surface quickly with a clean dry cloth or filter paper, and weighed each sample again to an accuracy of 0.1mg. The test results of three samples were taken as the arithmetic mean.

Water contact angle analysis. The test equipment was purchased from Kunshan xuncai instrument technology co., ltd (Kunshan, China). According to ISO15989:2004, the syringe capacity was 1 mL, the nominal diameter of stainless steel flat-headed needle was 0.52 mm, the water consumption was 1μL each time, the water drop image was magnified 25 times and projected on the screen, and the contact angle was measured within (60 ± 10) s after the water drop had transferred. The sample size was 25 mm × 300 mm, and the contact angle was measured 10 times on the same sample and averaged. Before the test, the sample was adjusted for 48 h at a temperature of (23 ± 2) °C and a relative humidity of (50 ± 5)%.

Friction and wear performance test. The friction and wear performance test referred to the “Test Method for Sliding Friction and Wear of Plastics”, GB/T3960-2016. The test equipment was purchased from Jinan Yihua Tribology Testing Technology Co., Ltd., Jihan, China. In the test, the sample remained stationary, the test ring rotated at 200 r/min, the test time was 30 min, and the load was 196 N. There were three samples in each group, the friction curve took the middle value, and the wear rate took the wear value corresponding to the friction curve.

Scanning electron microscopy analysis. The friction and wear surface of PAI material was observed by EVO 18 scanning electron microscope of Zeiss Company in Jena, Germany, and the surface morphology was studied to analyze the friction and wear mechanism. The worn surface was sprayed with gold, and then the test sample was fixed on the sample table by a conductive adhesive.

## 3. Results and Discussion

### 3.1. Structural Characterization of PAI

#### 3.1.1. Infrared Spectroscopy (FT-IR)

The structures of PAI(O), PAI-1, PAI-2, PAI-3, PAI-4, PAI-5, PAI(F) were characterized by infrared spectroscopy, and the results are shown in Figure 2. The characteristic peaks 1670 cm^−1^ and 1420 cm^−1^ corresponded to the stretching vibration absorption peak of amide bond C = O and the bending vibration peak of N-H, respectively. It indicated that all of the seven synthesized resins contained amide groups.

The characteristic peaks 1720 cm^−1^ and 1780 cm^−1^ were symmetric and asymmetric stretching vibration absorption peaks of imine structure -CO-N-CO-, 1380 cm^−1^ was stretching vibration absorption peak of C-N in imine, and 725 cm^−1^ was bending vibration absorption peak of imine -CO-N-CO- [16]. All of the seven synthesized resins contained these four characteristic absorption peaks, indicating that the synthesized resins all had an imide ring structure. In addition, there were no obvious amino characteristic peaks (amino stretching vibration peak 3254 cm^−1^, asymmetric stretching vibration peak 3194 cm^−1^) and acid anhydride characteristic peaks (C = O stretching vibration peak 1820 cm^−1^) in the spectrograms, which indicated that amino functional groups and acid anhydride functional groups reacted fully in the synthesis process.

In addition, all PAI molecular structures contained aromatic ether bonds, so there were corresponding characteristic absorption peaks at 1010 cm^−1^ (weak) and 1230 cm^−1^ (strong).

#### 3.1.2. Measurement of Molecular Weight and Polydispersity Index

The data in Table 1 showed that PAI(O), PAI-1, PAI-2, PAI-3, PAI-4, PAI-5, and PAI(F) all had relatively high molecular weights, and their molecular weights were similar, with no difference in the order of magnitude. The similar molecular weight could minimize the influence of molecular weight on the mechanical properties, friction and wear resistance, heat resistance, and other properties of materials, thus laying a foundation for the systematic study of the influence of side-chain trifluoromethyl on the properties of materials.

### 3.2. Heat Resistance Test

#### 3.2.1. DSC Test Characterization

PAI DSC curves and glass transition temperature values are shown in Figure 3 and Table 2, respectively.The glass transition temperature of the PAI polymer synthesized with 4,4’- diaminodiphenyl ether was 266.5 °C. With the increase of the number of 2,2’-bis (trifluoromethyl)-4,4’-diaminophenyl ether, the glass transition temperature of PAI terpolymer gradually increased. The data showed that the two trifluoromethyl groups in PAI side chain increased the steric effects and hindered the thermal movement of the chain segment. When the molar ratio of two diamines exceeded 5:5, the Tg of PAI terpolymer had little change.

#### 3.2.2. TGA Test Characterization

PAI TGA curves and Td_5%_ temperature values are shown in Figure 4 and Table 3, respectively. The Td_5%_ of PAI copolymer synthesized by 4,4’-diaminodiphenyl ether and TMAC was 476.5 °C. With the increase of the number of 2,2’-bis (trifluoromethyl)-4,4’-diaminophenyl ether, the Td_5%_ of PAI terpolymer gradually decreased from 472.1 °C to 461.3 °C. The main chain structure of 2,2’-bis (trifluoromethyl)-4,4’-diaminophenyl ether was the same as that of 4,4’-diaminodiphenyl ether. The results show that the trifluoromethyl in the side chain would reduce Td_5%_, which might be due to the larger volume of trifluoromethyl, which increased the distance between molecular chains and segments, thus reducing the intermolecular force and the packing density of PAI molecular chains, thus reducing the heat resistance of the material. The minimum Td_5%_ of PAI copolymer synthesized by 2,2’-bis (trifluoromethyl)-4,4’-diaminophenyl ether and TMAC is 459.6 °C.

### 3.3. Tensile Property Test

The tensile strength and elongation at break of PAI materials are shown in Table 4. The PAI copolymer synthesized by 4,4’-diaminodiphenyl ether and TMAC had a tensile strength of 133.4 MPa and an elongation at break of 9.46%. With the increase of the number of 2,2’-bis (trifluoromethyl)-4,4’-diaminophenyl ether, the tensile strength and elongation at break of PAI terpolymer decreased gradually.

This might be due to the large volume of trifluoromethyl, which reduced the force between PAI molecular chains and the bulk density, which led to the decrease of the tensile strength of the material with the increase of the amount of 2,2’-bis (trifluoromethyl)-4,4’-diaminophenyl ether. In addition, the steric hindrance of the two trifluoromethyl groups in the side chain was large, which made the molecular chain slip more difficult when the material was stretched. Therefore, the elongation at break decreased with the increase of the amount of 2,2’-bis (trifluoromethyl)-4,4’-diaminophenyl ether.

The tensile strength and elongation at break of PAI copolymer synthesized by 2,2’-bis (trifluoromethyl)-4,4’-diaminophenyl ether reached the minimum value of 61.3 MPa and 3.86%, respectively.

### 3.4. Characterization of Friction and Wear Properties

#### 3.4.1. Friction Coefficient and Wear Test

The friction-time curve and wear data are shown in Figure 5 and Table 5, respectively. The friction coefficient of PAI polymer synthesized by 4,4’-diaminodiphenyl ether was 0.4471. With the increase of the number of 2,2’-bis (trifluoromethyl)-4,4’-diaminophenyl ether monomer, the friction coefficient of PAI terpolymer changed little. When the molar ratio of 4,4’-diaminodiphenyl ether to 2,2’-bis (trifluoromethyl)-4,4’-diaminophenyl ether was 9: 1, the friction coefficient of PAI terpolymer increased slightly, reaching 0.4604. When all diamine monomers used were 2,2’-bis (trifluoromethyl)-4,4’-diaminophenyl ether, the friction coefficient of PAI reached the maximum value of 0.4897.

The results showed that when the number of2,2′-bis (trifluoromethyl)-4,4′-diaminophenyl ether was relatively low, the influence of trifluoromethyl on PAI molecular chain was relatively small. When the amount of 2,2′-bis (trifluoromethyl)-4,4′-diaminophenyl ether reached a certain level, trifluoromethyl would increase the friction coefficient of PAI material. This was because the volume of trifluoromethyl in the side chain was relatively large, and the steric effect was obvious when the content was relatively high, which would affect the regularity and flexibility of the PAI molecular chain and increase the friction coefficient [17,18]. This was consistent with the research report that the friction coefficient gradually increased with the decrease of flexibility [19,20]. The wear of PAI(O), PAI-1, PAI-2, PAI-3, PAI-4, PAI-5, and PAI(F) gradually increased. This was because the volume of trifluoromethyl was relatively large. With the increase of trifluoromethyl content, the intermolecular force of PAI decreased, and the bulk density of PAI also decreased. When the material was subjected to friction shear, the material loss easily occurred on the surface.

#### 3.4.2. Scanning Electron Microscope Analysis

The friction surfaces of PAI material were scanned by electron microscope, as shown in Figure 6. The wear mechanism of PAI copolymer synthesized by 4,4′-diaminodiphenyl ether was abrasive wear and slight adhesive wear. With the increase of the amount of 2,2′-bis (trifluoromethyl)-4,4′-diaminophenyl ether, the adhesive wear degree of PAI terpolymer increased. When the molar ratio of 4,4′-diaminodiphenyl ether to 2,2′-bis (trifluoromethyl)-4,4′-diaminophenyl ether was 5:5, the adhesive wear degree of PAI terpolymer was the highest. When the molar ratio of 4,4′-diaminodiphenyl ether to 2,2′-bis (trifluoromethyl)-4,4′-diaminophenyl ether was 9:1, the wear mechanism of PAI terpolymer was abrasive wear. The wear mechanism of PAI copolymer synthesized by 2,2′-bis (trifluoromethyl)-4,4′-diaminophenyl ether was mainly abrasive wear. The results showed that the wear mechanism of PAI material changed with the different amount of diamine monomer.

### 3.5. Hydrophobic Performance Test

The water contact anglediagrams of PAI are shown in Figure 7, and the specific values are shown in Table 6. The water contact angle of PAI synthesized by 4,4′-diaminodiphenyl ether was 77.7°. With the increase of the amount of2,2′-bis (trifluoromethyl)-4,4′-diaminophenylether monomer, the water contact angle of PAI material gradually increased. The water contact angle of PAI synthesized by 2,2′-bis (trifluoromethyl)-4,4′-diaminophenyl ether was the highest, reaching 92.3°. The larger the water contact angle, the stronger the hydrophobicity of the material surface. The results showed that the hydrophobic ability of PAI increased with the increase of trifluoromethyl content, which was due to the hydrophobicity of fluorine atoms and the hydrophobicity of PAI enhanced by fluorine-containing groups [7,8]. The water absorption of PAI(O), PAI-1, PAI-2, PAI-3, PAI-4, PAI-5, and PAI(F) decreased with the increase of trifluoromethyl content. The research results of water absorption also showed that trifluoromethyl could reduce the water absorption of materials [7,8].

## 4. Conclusions

In this paper, 4,4’-diaminodiphenyl ether and 2,2’-bis (trifluoromethyl)-4,4’-diaminophenyl ether were selected for molecular structure design, and PAI materials were prepared by acyl chloride method, and the effects of Trifluoromethyl group ratios on the properties of PAI materials were studied. The main research conclusions were as follows:(1)The Tg of PAI polymer synthesized with 4,4’-diaminodiphenyl ether as diamine monomer was 266.5 °C. With the increase of the amount of 2,2’-bis (trifluoromethyl)-4,4’-diaminophenyl ether, the glass transition temperature of PAI terpolymer gradually increased. When the molar ratio of the two diamines was 5:5, the Tg of PAI terpolymer reached 282.9 °C, and then with the increase of the amount of 2,2’-bis (trifluoromethyl)-4,4’-diaminophenyl ether, the Tg changed little.(2)The Td_5%_ of PAI copolymer synthesized by 4,4’-diaminodiphenyl ether and TMAC was 476.5 °C. With the increase of the dosage of 2,2’-bis (trifluoromethyl)-4,4’-diaminophenyl ether, the Td_5%_ of PAI material gradually decreased. The Td_5%_ of PAI copolymer synthesized by 2,2’-bis (trifluoromethyl)-4,4’-diaminophenyl ether reached the minimum value of 459.6 °C.(3)The tensile strength and elongation at break of PAI synthesized by 4,4’-diaminodiphenyl ether were the highest, which were 133.4 MPa and 9.46% respectively. With the increase of the dosage of 2,2’-bis (trifluoromethyl)-4,4’-diaminophenyl ether, the tensile strength and elongation at break of PAI materials decreased gradually. The tensile strength and elongation at break of PAI copolymer synthesized by 2,2’-bis (trifluoromethyl)-4,4’-diaminophenyl ether reached the minimum value of 61.3 MPa and 3.86%, respectively.(4)The wear mechanism of PAI varied with the content of trifluoromethyl. With the increase of the amount of fluorinated diamine monomer, the adhesive wear degree of PAI materials gradually increased, and reached the maximum when the molar ratio of the two monomers was 5:5, and then decreased gradually. Different trifluoromethyl content had little effect on friction coefficient, and the friction coefficient increased slightly when the molar ratio of 4,4 ′-diaminodiphenyl ether to 2,2′-bis (trifluoromethyl)-4,4 ′-diaminophenyl ether is 1:9. With the increase of trifluoromethyl content, the wear of PAI material would increase.(5)PAI material synthesized by 4,4’-diaminodiphenyl ether had the smallest water contact angle and the largest water absorption rate. With the increase of the amount of 2,2’-bis (trifluoromethyl)-4,4’-diaminophenyl ether, the water contact angle of PAI material increased and the water absorption decreased. The results showed that trifluoromethyl had good hydrophobic properties.

To sum up, the results of this study showed that the introduction of trifluoromethyl into the side chain provided an effective way to prepare PAI materials with low water absorption. Considering the comprehensive properties such as heat resistance, friction and wear, tensile properties, etc., the appropriate addition amount was 10–30%.
**PAI materials synthesized by 4,4′-diaminodiphenyl ether/2,2′-bis (trifluoromethyl)-4,4′-diaminophenyl ether and their properties**Haiyang Yang, Duxin Li*, Jun Yang*, Jin Wang, Shunchang GanIn this paper, 4,4′-diaminodiphenyl ether and 2,2′-bis (trifluoromethyl)-4,4′-diaminophenyl ether are selected for molecular structure design, and PAI materials are synthesized by acyl chloride method. The results of this study showed that the introduction of trifluoromethyl into the side chain provided an effective way to prepare PAI materials with low water absorption. Considering the comprehensive properties such as heat resistance, friction and wear, tensile properties, etc., the appropriate addition amount is 10–30%.
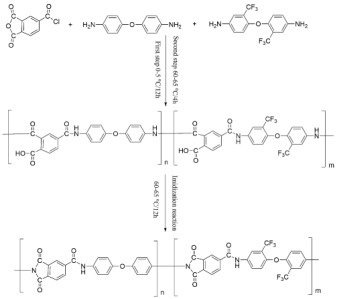


## Data Availability

The data obtained in this study are authentic and not supported by other reports.

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
