# Peer review of "PAI Materials Synthesized by 4,4′-Diaminodiphenyl ether/2,2′-Bis (trifluoromethyl)-4,4′-diaminophenyl ether and Their Properties"

_materials, 2021, doi:10.3390/ma14216376_

Round 1
Reviewer 1 Report
This paper is interesting in terms of materials, synthesis, characterization, and applications. Nevertheless, it needs substantial improvement for details and presentation. Moreover, written English MUST be improved (whole document: seek help). This reviewer recommends this manuscript be published after major revisions to add additional rigor and clarity. Here are some comments to address: 1. For the whole manuscript, Tg and Td5% should be Tg and Td5%. 2. There is no more information in the introduction part and should be re-edited and written with the relevant information. 3. Line 99, 20 wt% PAI solution. 4. Line 58, 4,4'-diaminodiphenyl ether was analytically pure and purchased from Alad- din (Shanghai China). 5. The material section thoroughly checks English and writing. There is no space between each word. 6. Line 76, 500mL should be 500 mL 7. Line 78,79, 81,85: There is no space between each word, for instance, 0.05mol. It should be 0.05 mol. 8. Revise the manuscript completely. There are a lot of mistakes in this whole manuscript, which means there is no space between every word and unit/parameter. 9. Range/parameters, sample weight, etc. should be given in more detail in the characterization part (section 2.3). 10. The structures of PI cannot be confirmed by FTIR analysis alone. 1H NMR spectrum analysis should be provided. 11. The DSC analysis shows that all PAIs are amorphous, while there is no evidence that the Tg peak appeared in the DSC traces. Therefore, DMA analysis should be provided to clearly detect the Tg values. 12. Authors should provide images of the tensile test. 13. One decimal is enough for water contact angle values.Author Response
Dear reviewer:
Hello, thank you for your hard work. The reply to your comments and suggestions is shown in the annex. If you have any new comments or suggestions, please give me your advice.
- For the whole manuscript, Tg and Td5% should be Tg and Td5%.
Comments reply:The wrong writing format has been corrected.
- There is no more information in the introduction part and should be re-edited and written with the relevant information.
Comments reply:Optimization will be carried out according to the opinions and suggestions of reviewers.
- Line 99, 20 wt% PAI solution.
Comments reply:20wt.% means the mass concentration is 20%.
- Line 58, 4,4'-diaminodiphenyl ether was analytically pure and purchased from Alad- din (Shanghai China).
Comments reply:It has been corrected.
- The material section thoroughly checks English and writing. There is no space between each word.
Comments reply:The problematic part has been corrected.
- Line 76, 500mL should be 500 mL
Comments reply:It has been corrected.
- Line 78,79, 81,85: There is no space between each word, for instance, 0.05mol. It should be 0.05 mol.
Comments reply:It has been corrected.
- Revise the manuscript completely. There are a lot of mistakes in this whole manuscript, which means there is no space between every word and unit/parameter.
Comments reply:They have been corrected.
- Range/parameters, sample weight, etc. should be given in more detail in the characterization part (section 2.3).
Comments reply:The standards used in the test method have been marked in the 2.3 characterization section. If any standards need to be described in more detail, If the reviewing teacher needs some more details, and I will improve the specific details.
- The structures of PI cannot be confirmed by FTIR analys is alone. 1H NMR spectrum analysis should be provided.
Comments reply:The diamine monomer and acyl chloride used in the synthesis of PAI in this paper are mature industrial products, and the synthesis method is also the conventional synthesis method. Infrared spectrum is only evidence, which shows that the synthesized polymer contains amide bond and imide ring. The emphasis of this paper is to study the influence of trifluoromethyl content on the properties of materials. In addition, terpolymers will interfere with the analysis of NMR spectra.
- The DSC analysis shows that all PAIs are amorphous, while there is no evidence that the Tg peak appeared in the DSC traces. Therefore, DMA analysis should be provided to clearly detect the Tg values.
Comments reply:DSC and DMA are used to test the glass transition temperature of materials. There is a step in DSC curve and a peak in DMA curve. In this paper, one of the temperature plots is selected, and the step of glass transition temperature is obvious on DSC testing instrument. Therefore, in this paper, DSC was chosen to test the glass transition temperature, and DMA was not used to test the glass transition temperature.
- Authors should provide images of the tensile test.
Comments reply:The number of test splines was 5, and the arithmetic average was taken. The tensile curve of each sample is different.
- One decimal is enough for water contact angle values.
Comments reply:They have been corrected.
Haiyang Yang

Reviewer 2 Report
In the submitted manuscript, the authors describe the results of their investigations on the effect of trifluoromethyl substation in a series of poly(amide-imide) copolymers obtaind by step-growth polymerization of 4'-diaminodiphenyl ether and 2,2'-bis (trifluoromethyl)-4,4'-diamino-phenyl ether. It is proposed to accept this manuscript for publication after some revision on the basis of comments below.
In the Introduction, the structure of PAI should be deleted from the text, and should be displayed as a separate scheme, that is as Scheme 1.
Pages 3-4, lines 116-122, the temperature of the GPC measurement should be provided.
It would be more informative to magnify the 500-2000 cm(-1) region in the the IR spectra in Figure 2 instead of showing a much broader range in which there are no any signals.
In Table 1, the “distribution index” expression is not correct. This is called “polydispersity index”, and the authors should correct this.
It would be helpful for the readers to indicate the glass transition temperatures with small arrows in Figure 3.
In the SEM images in Figure 6, the magnification is quite low, on the one hand. At the presented magnification, there are no substantial differences between the SEM pictures, on the other hand. Therefore, it is absolutely unclear what the authors discuss in Section 3.4.2. Scanning electron microscope analysis (lines 266-280). The authors should explain in details their conclusion in this section of their manuscript.
Author Response
Dear reviewer:
Hello, thank you for your hard work. The reply to your comments and suggestions is shown in the annex. If you have any new comments or suggestions, please give me your advice.
1、In the Introduction, the structure of PAI should be deleted from the text, and should be displayed as a separate scheme, that is as Scheme 1.
Review reply:It has been corrected.
2、Pages 3-4, lines 116-122, the temperature of the GPC measurement should be provided.
Review reply:The ambient temperature of the test is 23±1℃. This clause of test temperature has been added to the test standard.
3、It would be more informative to magnify the 500-2000 cm(-1) region in the the IR spectra in Figure 2 instead of showing a much broader range in which there are no any signals.
Review reply:Infrared spectrum only proves that the synthesized material contains amide bond and imide structure. Therefore, the characteristic absorption peaks of amide bond and imide ring are highlighted. The two characteristic peaks of aromatic ether bond are also obvious, so they are also identified.
4、In Table 1, the “distribution index” expression is not correct. This is called “polydispersity index”, and the authors should correct this.
Review reply:It has been corrected.
5、It would be helpful for the readers to indicate the glass transition temperatures with small arrows in Figure 3.
Review reply:The arrow has been identified in fig. 3.
6、In the SEM images in Figure 6, the magnification is quite low, on the one hand. At the presented magnification, there are no substantial differences between the SEM pictures, on the other hand. Therefore, it is absolutely unclear what the authors discuss in Section 3.4.2. Scanning electron microscope analysis (lines 266-280). The authors should explain in details their conclusion in this section of their manuscript.
Review reply:Some contents have been optimized.
When the ratio of two diamine monomers is before 9: 1, the wear mechanism of PAI is abrasive wear and slight adhesive wear. When the ratio of two diamine monomers is above 9: 1, the wear mechanism of the material is mainly abrasive wear.
Haiyang Yang
54180283@qq.com

Round 2
Reviewer 1 Report
The authors have done their revision works as per the recommendation of the reviewer. Now, it is suitable to publish this manuscript.